# Risk factors associated with poor pain outcomes following primary knee replacement surgery: Analysis of data from the clinical practice research datalink, hospital episode statistics and patient reported outcomes as part of the STAR research programme

Hasan Raza Mohammad[1], Rachael Gooberman-Hill[2,3], Antonella Delmestri[1,4], John Broomfield[1], Rita Patel[2], Joerg Huber[5], Cesar Garriga[1,6], Christopher Eccleston[7], Rafael Pinedo-Villanueva[1], Tamer T. Malak[1], Nigel Arden[1], Andrew Price[1], Vikki Wylde[2,3], Tim J. Peters[8], Ashley W. Blom[2,3], Andrew Judge[1,2,3,4]*

1 Nuffield Department of Orthopaedics, Rheumatology and Musculoskeletal Sciences, Nuffield Orthopaedic Centre, University of Oxford, Oxford, United Kingdom, 2 Musculoskeletal Research Unit, Translational Health Sciences, Bristol Medical School, University of Bristol, Learning and Research Building, Bristol, United Kingdom, 3 National Institute for Health Research Bristol Biomedical Research Centre, University Hospitals Bristol and Weston NHS Foundation Trust and University of Bristol, Bristol, United Kingdom, 4 Nuffield Department of Orthopaedics, Centre for Statistics in Medicine, Rheumatology and Musculoskeletal Sciences, Nuffield Orthopaedic Centre, University of Oxford, Oxford, United Kingdom, 5 Department of Orthopedics, Stadtspital Triemli, Zurich, Switzerland, 6 Nuffield Department of Primary Care Health Sciences, University of Oxford, Oxford, United Kingdom, 7 Department for Health, Centre for Pain Research, University of Bath, Bath, United Kingdom, 8 Population Health Sciences, Bristol Medical School, University of Bristol, Learning and Research Building, Bristol, United Kingdom

* Andrew.judge@bristol.ac.uk

## Abstract

### Objective

Identify risk factors for poor pain outcomes six months after primary knee replacement surgery.

### Methods

Observational cohort study on patients receiving primary knee replacement from the UK Clinical Practice Research Datalink, Hospital Episode Statistics and Patient Reported Outcomes. A wide range of variables routinely collected in primary and secondary care were identified as potential predictors of worsening or only minor improvement in pain, based on the Oxford Knee Score pain subscale. Results are presented as relative risk ratios and adjusted risk differences (ARD) by fitting a generalized linear model with a binomial error structure and log link function.

**Data Availability Statement:** Electronic health records are, by definition, considered sensitive data in the UK by the Data Protection Act and cannot be shared via public deposition because of information governance restriction in place to protect patient confidentiality. Access to data is available once approval has been obtained through the individual constituent entities controlling access to the data. The primary care data can be requested via application to the Clinical Practice Research Datalink, secondary care data can be requested via application to the Hospital Episode Statistics from the UK Health and Social Care Information Centre, and mortality data are available by application to the UK Office for National Statistics. Alternatively, access to secondary care data and mortality data is available through linkage requested upon application for primary care data to the Clinical Practice Research Datalink. Information regarding linkage is available from: https://www.cprd.com/linked-data (https://www.cprd.com/research-applications).

**Funding:** Funding This study is funded by the National Institute for Health Research (NIHR) [Programme Grant for Applied Research (Grant Reference Number RP-PG-0613-20001)]. This study was also supported by the NIHR Biomedical Research Centre at the University Hospitals Bristol and Weston NHS Foundation Trust and the University of Bristol (Grant Reference number: IS-BRC-1215-20011). The views expressed are those of the authors and not necessarily those of the NIHR or the Department of Health and Social Care. Disclaimer The views expressed represent those of the authors and do not necessarily reflect those of the NHS, the National Institute for Health Research, the Programme Grants for Applied Research (PGfAR) Programme, the Department of Health and Social Care or the Healthcare Quality Improvement Partnership (HQIP) who do not vouch for how the information is presented.

**Competing interests:** All authors have completed the Unified Competing Interest form at www.icmje.org/coi_disclosure.pdf and declare: AJ reports grants from NIHR PGfAR, during the conduct of the study; personal fees from Freshfields Bruckhaus Deringer, personal fees from Anthera Pharmaceuticals Ltd, outside the submitted work; RGH reports grants from NIHR PGfAR, during the conduct of the study. HRM reports grants from Royal College of Surgeons Research Fellowship, outside the submitted work; RPV reports research funding from UK-NIHR, Kyowa Kirin Services, International Osteoporosis Foundation, and lecture fees and/or consulting honoraria from Amgen, UCB, Kyowa Kirin Services, and Mereo Biopharma,

## Results

Information was available for 4,750 patients from 2009 to 2016, with a mean age of 69, of whom 56.1% were female. 10.4% of patients had poor pain outcomes. The strongest effects were seen for pre-operative factors: mild knee pain symptoms at the time of surgery (ARD 18.2% (95% Confidence Interval 13.6, 22.8), smoking 12.0% (95% CI:7.3, 16.6), living in the most deprived areas 5.6% (95% CI:2.3, 9.0) and obesity class II 6.3% (95% CI:3.0, 9.7). Important risk factors with more moderate effects included a history of previous knee arthroscopy surgery 4.6% (95% CI:2.5, 6.6), and use of opioids 3.4% (95% CI:1.4, 5.3) within three months after surgery. Those patients with worsening pain state change had more complications by 3 months (11.8% among those in a worse pain state vs. 2.7% with the same pain state).

## Conclusions

We quantified the relative importance of individual risk factors including mild pre-operative pain, smoking, deprivation, obesity and opioid use in terms of the absolute proportions of patients achieving poor pain outcomes. These findings will support development of interventions to reduce the numbers of patients who have poor pain outcomes.

## Introduction

Knee replacement surgery may be offered to patients with knee osteoarthritis who have not responded to conservative treatment [1]. Over 100,000 knee replacement operations are carried out each year in the UK for osteoarthritis and other surgical indications [2]. Many patients can expect to achieve reductions in knee pain and improvements in functional outcomes following surgery [3]. The percentage who experience ongoing chronic knee pain post-surgery is variable [4], with up to 20% experiencing knee pain that impacts their quality of life after 3 months post-op [5]. Patients who experience this kind of pain after surgery have not received the expected benefit and for some their pain is worse than it was before the operation [6, 7].

It is important to identify which patients are at greatest risk of similar or worse pain after knee replacement. When healthcare professionals and patients are making decisions about treatment options, knowledge of the chances of benefit and the risk of harms, or of no benefit, is key to informed choice. Although previous research has explored predictors of outcomes of knee replacement [8], most studies have focused on total scores encompassing several domains (e.g. pain, stiffness and function) and fewer studies have focussed solely on pain status [7]. Most existing studies tend to use continuous patient-reported outcome scores, but as the majority of patients achieve good outcomes, they can only help us identify predictors that differentiate between patients that achieve a 'really good outcome' versus a 'good outcome' [9, 10]. Instead, defining pain outcome by focussing on those patients whose symptoms have had no clinically meaningful change or which worsened after surgery, would allow identification of poor outcomes [11–14]. The predictive ability of previous studies is poor, which means that they are unable to explain variation in patient-reported pain after knee replacement [7, 10]. Most research has explored pre-operative risk factors, but little research has sought to understand a wider range of pain determinants that occur post-operatively. These are important because the time after surgery may present an ideal opportunity for targeted intervention to prevent the persistence or worsening of pain.

all outside of the scope of this study; NA reports personal fees from Pfizer/Lilly, personal fees from Bristows LLP, grants from Merck Grant, outside the submitted work; VW reports grants from NIHR, during the conduct of the study; TJP reports grants from UK NIHR Programme Grant for Applied Research, during the conduct of the study. All other authors declare no conflicts of interest. This does not alter our adherence to PLOS ONE policies on sharing data and materials.

The aim of this study is to identify pre- and post-operative risk factors for whether or not a patient has a poor pain outcome after knee replacement surgery, by analysing a wide range of potential factors from the UK Clinical Practice Research Datalink (CPRD) primary care GOLD database linked to English Hospital Episode Statistics (HES) hospital admissions and to Patient Reported Outcomes Measures (PROMs) data.

## Methods

### Study design

Retrospective observational study using anonymised linked data from CPRD, HES and PROMs.

### Data source

CPRD GOLD contains anonymised individual patient data from electronic primary healthcare records from practices across the United Kingdom [15]. CPRD is one of the largest databases of longitudinal primary care medical records in the world with coverage of 674 general practices in the UK with 11.3 million patients, of which 4.4 million patients are active [15]. Primary care records from CPRD were linked to secondary care admission records from HES Admitted Patient Care data and to Office for National Statistics (ONS) mortality data. From 1st April 2009, HES provides PROMs data before and six months following knee replacements. Linkage of CPRD-HES-ONS-PROMS data is done by NHS Digital as a 'trusted third party'.

### Sample

We included all patients receiving a primary total or uni-compartmental knee replacement between 2009 and 2016. Inclusion in the analysis was limited to those patients with HES linked data (England only) who completed both the pre- and six-month post-operative Oxford Knee Score pain subscale (see Flow Diagram Fig 1).

### Main outcome measure

The Oxford Knee score (OKS) [16] is collected as part of the national PROMs programme and is a measure of patient-reported pain and function. Each of the 12 questions is scored between 0 (meaning worst symptoms) and 4 (least troublesome symptoms). Pain- and function-related subscales within the OKS have been identified and validated [17]. An OKS pain subscale (OKS-PS) summary score can be calculated, ranging from 0 (most pain) to 28 (least pain), by summing the responses of the seven OKS-PS items.

The Treatment Effect (TE = (pre-treatment score − post treatment score)/pre-treatment score) [12, 18] was calculated for each patient using the OKS-PS score (normalized to a score from 0 (least pain) to 100 (worst pain)). A TE of 1 (best score) corresponds to a patient without pain after treatment, a TE of 0 to no improvement and no deterioration, a negative TE to more pain at follow up. TE ≤0.2 is used to classify whether or not a patient responds to surgery in respect of their knee pain, and has previously been validated against the OARSI-OMERACT criteria [14] to identify responders to surgery [12].

### Predictor variables

In discussion with clinicians we sought to identify variables within this routinely collected dataset which may provide a wide range of potential predictors for the pain outcome.

**Pre-operative predictors.** To measure socioeconomic deprivation, we used the index of multiple deprivation (IMD) score. This is a relative measure of deprivation for small areas—

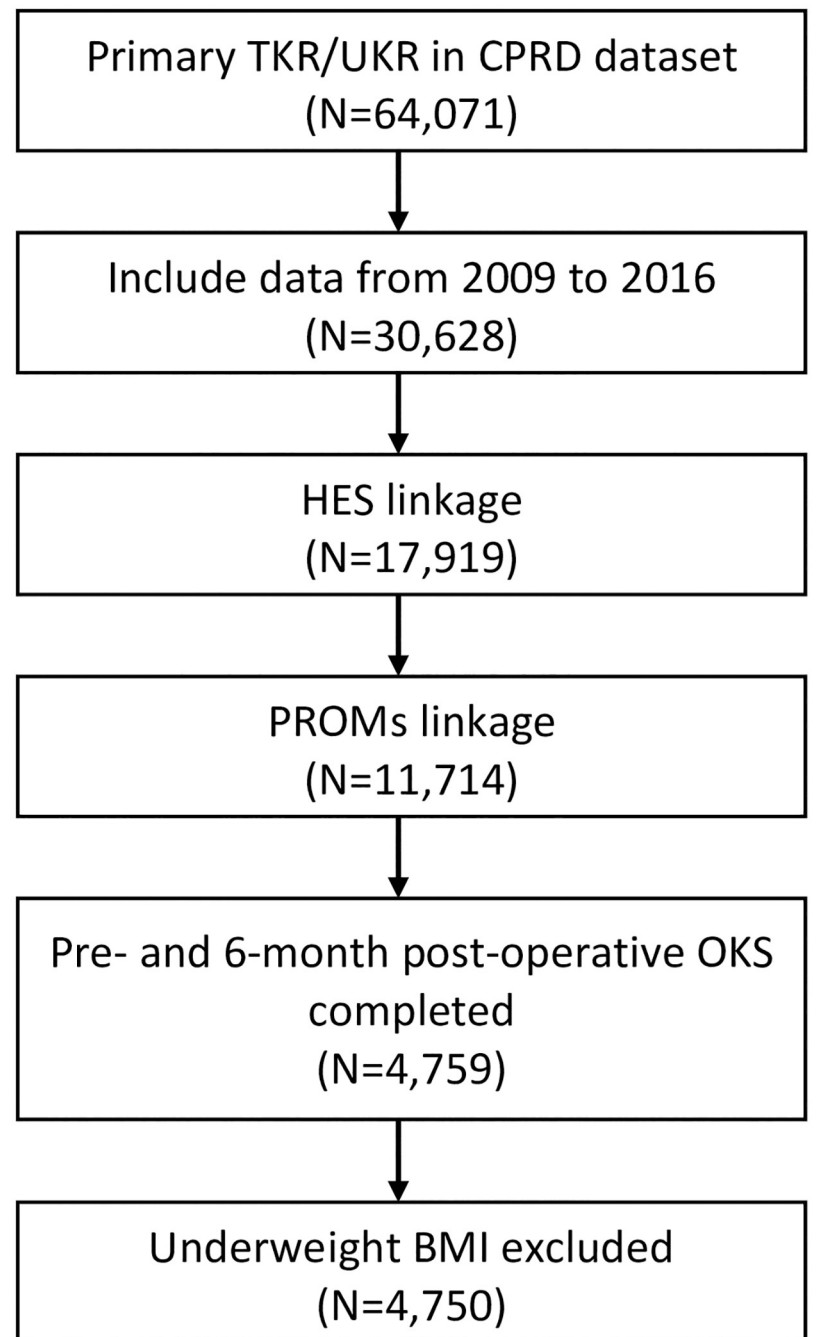

**Fig 1. Patient flow diagram.** TKR/UKR, total and uni-compartmental knee replacement; CPRD, Clinical Practice Research Datalink GOLD; HES, English Hospital Episode Statistics; PROMs, Patient Reported Outcome Measures; OKS, Oxford Knee Score; Underweight BMI, Body Mass Index under 18.5 Kg/m$^2$.

termed lower super output areas (LSOAs)–which are defined as geographical areas of a similar population size, with an average of 1,500 residents [19]. The IMD comprises seven measures of deprivation: income deprivation; employment deprivation; education, skills and training deprivation; health deprivation and disability; crime; barriers to housing and services; and living environment deprivation. We used the IMD rank for a patient's LSOA and categorised

patients into quintiles based upon the national ranking of local areas, with quintile 1 being the least deprived group and quintile 5 being the most deprived group (*i.e.* reordered to aid reporting). As a measure of comorbidity we used the Royal College of Surgeons' (RCS) Charlson Score, which is calculated based on the presence of several chronic conditions identified using ICD-10 codes at the time of knee replacement surgery admission and all admissions in the preceding 5 years [20].

Patient case-mix factors included: pre-operative OKS pain score, age at surgery, Body Mass Index (BMI), smoking, alcohol consumption, gender, index of multiple deprivation (IMD) score, Charlson Comorbidity Index (five years prior to surgery). Previous medication use included steroids (non-glucocorticosteroids (non-GCS), Steroid (any type of injection), oral), non-steroidal anti-inflammatory drugs (NSAIDS), opioids, antibiotics, and antidepressants. We identified whether the primary procedure was a total or uni-compartmental knee replacement. We classified patients according to whether or not they had a knee arthroscopy prior to surgery.

**Post-operative predictors.**   Length of stay (LOS) at hospital was calculated as the number of days between the hospital admission and discharge date. We identified medical complications as one or more events happening 3 months after the operation from the following list: stroke (excluding mini stroke), respiratory infection, acute myocardial infarction, pulmonary embolism/deep vein thrombosis, urinary tract infection, wound disruption, surgical site infection, fracture after implant, complication of prosthesis, neurovascular injury, acute renal failure and blood transfusion [21].

Re-operations include stiffness requiring manipulation under anaesthetic (MUA), arthroscopic surgery, debridement for infection and operations for wound problems. Re-operations included open operations (such as debridements for infection or ligament repairs), arthroscopic operations (excision of loose bodies or menisci in uni-compartmental knee replacement (UKR), washouts/debridements for infection), and closed operations. We also evaluated the rate of revision by three months after the surgery.

We identified medications prescribed (including opioids, NSAIDS, and antibiotics) pre- and post-surgery and calculated the total number of general practice visits between surgery and 3 months post-surgery. We have assessed for evidence of collinearity using variance inflation factors and there was no evidence of multicollinearity.

## Statistical analysis

To describe their change in pain state before and 6 months after surgery, patients are categorised into pain groups, and descriptive statistics (without statistical testing) used to describe the number (percentage) of patients that move between different pain states before and after surgery. We then describe the characteristics of patients who are most likely to be moving to a worse pain state post-operatively.

Logistic regression modeling was used to describe the association of predictor variables with the outcome of interest (responder to surgery according to TE pain score). As our dataset is large, we selected the lowest category for each variable as the reference category. Composite variables were used if individual characteristics were rare. Results of the regression model are presented as relative risk ratios by fitting a generalized linear model with a binomial error structure and a log link function (log-logistic model). Results are further presented as adjusted risk differences estimated from marginal effects from the logistic regression model [22]. Fractional polynomial regression was used to assess evidence of linearity of continuous predictors with the outcome. As there was evidence of non-linearity for the pre-operative OKS pain score and BMI, these variables were categorized. We excluded nine underweight BMI patients, as

there were too few patients in this category, leading to a small cell problem in the multivariable regression model, and it would be inappropriate to combine underweight and normal BMI categories together. Multiple imputation by chained equations was used to account for the cumulative effect of missing data in several of the variables [23]. Forty imputed datasets were generated using all potential factors (including the outcome) and estimated parameters were combined using Rubin's rules. The C-statistic was used to describe the discriminatory ability of variables in the final model. We examine the strength of associations and not arbitrary measures of statistical significance with cut offs of for example $p<0.05$ or the related concept of whether the confidence interval includes the null value.

Analyses were conducted using Stata version 15.1 statistical software (StataCorp, College Station, Texas). We followed the STROBE (Strengthening the Reporting of Observational Studies in Epidemiology) guideline to report our study [24].

### Ethics approval

The CPRD Group has obtained ethical approval from a National Research Ethics Service Committee (NRES) for all purely observational research using anonymised CPRD data; namely, studies which do not include patient involvement. The study has been approved by ISAC (Independent Scientific Advisory Committee) for MHRA Database Research) (protocol number 11_050AMnA2RA2).

## Results

Information was available for 4,750 patients over the time period from 2009 to 2016 (Fig 1), with a mean age of 69 (SD 9), of whom 56.1% were female. Only four of the pre-operative variables had missing data: BMI (352, 7.4%), smoking (81, 1.7%), alcohol consumption (816, 17.2%) and IMD (5, 0.1%). Multiple imputation was used to account for missing data in the analysis. Assumptions of the imputation model were assessed and described in the (S1 Table).

Within the dataset, 10.4% of patients experienced a poor pain outcome following surgery, such that their symptoms of pain did not have a relative improvement of at least 20% or worsened by six months after surgery. Fig 2 shows the distribution of the treatment effects score; although the majority of patients had a good pain outcome (blue bars), there is a minority that have poor pain outcomes (red bars).

### Pre-operative predictors

The effect of pre-operative OKS-PS had a non-linear association with pain outcome (Fig 3). Patients with the mildest pain symptoms (OKS-PS 17–28) had a 3.9 times increased risk of a poor pain outcome compared to those with OKS-PS 8–10. A 'U-shaped' effect was observed for patient age at surgery, where those aged between 60–69 and 70–79 had a reduced risk of poor pain outcome compared with the youngest age group. The oldest age group was similar to the youngest. Compared with those of normal BMI, being overweight and obese increased risk of poor pain response to surgery. The highest proportion of patients with poor pain outcomes were in the group of current smokers, males, people living in the most deprived areas and those with inflammatory arthritis (Table 1). For the following variables there is weak evidence of an association. An effect of Charlson co-morbidities was only seen for those with four or more co-morbidities getting worse outcomes. Having a uni-compartmental rather than total knee replacement was associated with a better pain response, but this was attenuated when post-operative factors were included in the model. Patients who had previously had knee arthroscopy prior to their primary knee replacement were more likely to have poor pain

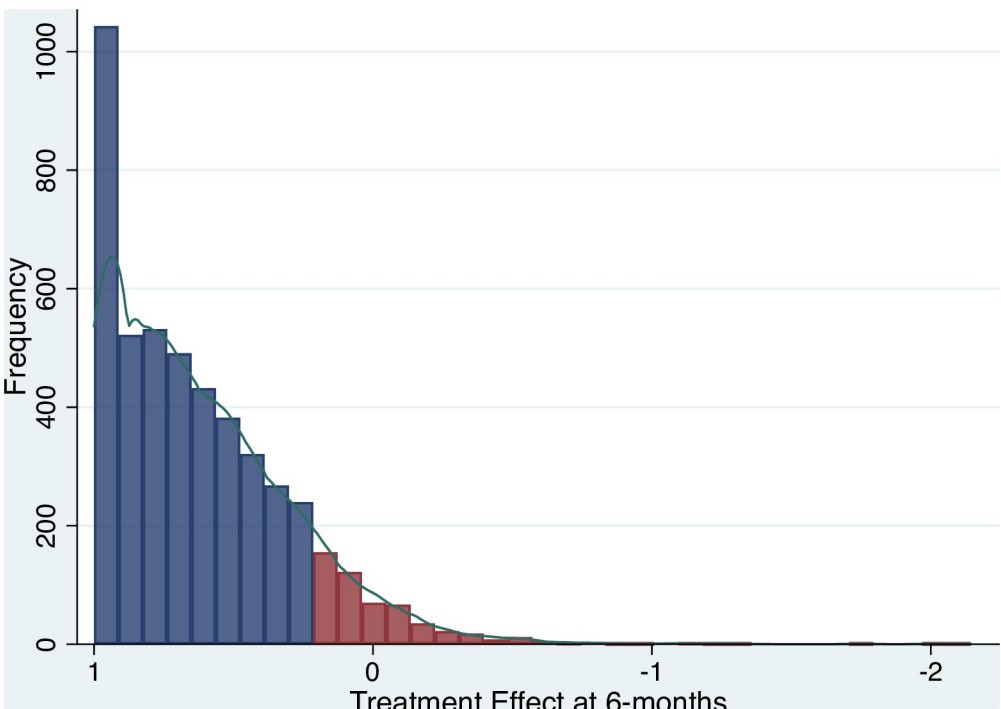

**Fig 2. Distribution of the treatment effect score for patients who did, and did not, respond to surgery.** Red = poor pain outcome, Blue = good pain outcome.

outcomes. Patients prescribed full opioids and antidepressants were more likely to have a poor pain response.

## Post-operative predictors

Medical complications occurring within three months of surgery were rare (Table 1) with an overall complication rate of 4.3%. The overall rate of re-operation was 3.0% and only eight (0.2%) patients were revised within three months. Re-admission to hospital for any reason after surgery was more common at 12.0%.

Re-admission to hospital, revision surgery, and manipulation under anaesthetic within three months of the operation were all associated with poor pain response to surgery at six months (Fig 3). In respect of medication use post operatively, patients prescribed opioids and antibiotics had a stronger association with poor pain response.

## Absolute risk differences

Table 1 presents descriptive statistics showing the observed proportions of poor outcomes for each category of the predictor variables. Fig 4 shows the adjusted risk differences. For the pre-operative OKS-PS, this can be interpreted as those having a score of 17–28 with a poor pain outcome 18.2 percentage points more often (95% Confidence Interval 13.6, 22.8) than with a pre-operative score of 8–10, on average. The other pre-operative variables with some absolute adjusted risk differences were smoking 12.0 percentage points (95% CI: 7.3, 16.6), obesity class II 6.3 percentage points (95% CI: 3.0, 9.7), and living in the most deprived areas 5.6 percentage points (95% CI: 2.3, 9.0). For the post-operative risk factors, revision surgery 15.8 percentage points (95% CI: -10.2, 41.7) and manipulation under anaesthetic 9.7 percentage points (95%

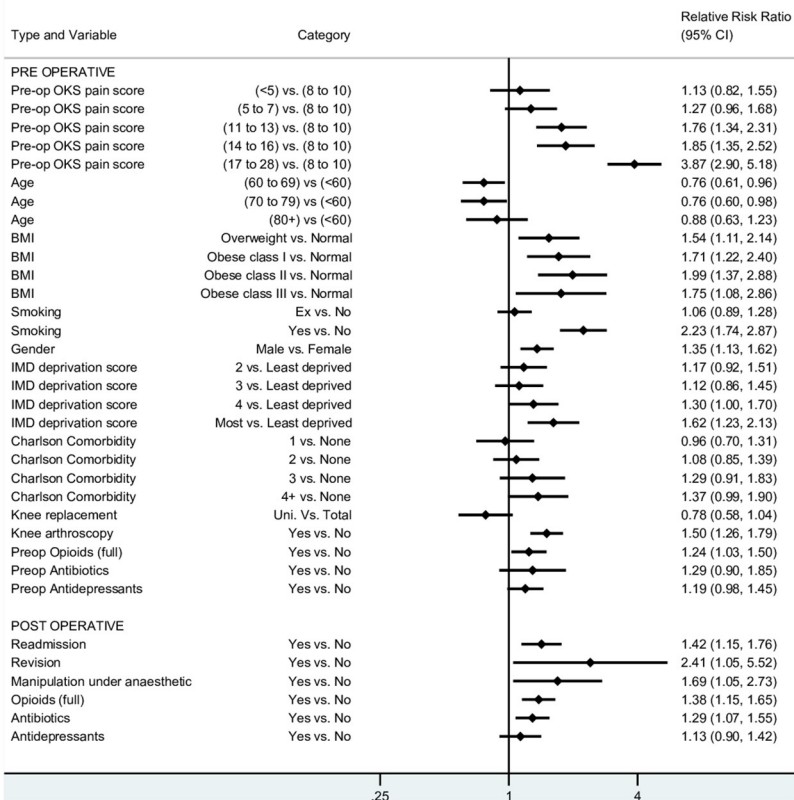

**Fig 3. Forest plot of predictors of poor pain outcomes.**

CI: -1.1, 20.5) conferred some absolute adjusted risk of poor pain outcome, although these were not statistically significant. Important risk factors with more moderate effects included a history of previous knee arthroscopy surgery 4.6 percentage points (95% CI:2.5, 6.6), and use of opioids 3.4 percentage points (95% CI:1.4, 5.3) within three months after surgery.

The discriminatory ability of variables in the final model was modest (c-statistic 0.72, 95% Confidence Interval 0.70, 0.75) (S1 Fig).

## Pain state change

Table 2 characterises patients' change in pain status before and at 6 months after surgery. Patients with the mildest pre-operative pain symptoms (OKS-PS 17–28) were most likely to not improve and move to a worse pain state following surgery, where 20% of patients had a poor pain outcome compared to around 10% in the other pre-operative pain states. To further understand why patients in this mild pre-operative pain state (OKS-PS 17–28) had worse pain outcomes, we describe the pre-operative and post-operative characteristics of those with worsening pain state change. These patients had more complications by 3 months (11.8% among those in a worse pain state vs. 2.7% with the same pain state); readmission (26.5% vs. 8.2%); and re-operation (5.9% vs. 3.0%). There were some differences in length of stay, where those who moved to a worse pain state had much shorter length of stay, particularly < 2-days (23.5% vs. 1.9%), with more pain medication use pre-operatively [(NSAIDS (94.1% vs. 85.0%), opioids (full) (35.3% vs. 25.6%); opioids (partial) (79.4% vs. 57.0%) and antibiotics (32.4% vs. 18.3%)]; and at 3 months post-operatively [(opioids (full) (50.0% vs. 22.1%), opioids (partial) (26.5% vs.

**Table 1. Descriptive statistics describing the total number of patients with each potential risk factor, and the proportion of patients with a poor pain outcome, according to whether or not they have the factor.**

| | Total | Proportion of patients with poor pain response with and without each risk factor | |
| --- | --- | --- | --- |
| | | Without | With |
| **PRE-OPERATIVE** | | | |
| Pre-op OKS pain score | | | |
| <5 | 562 (11.8%) | | 59 (10.5%) |
| 5 to 7 | 967 (20.4%) | | 97 (10.0%) |
| 8 to 10 | 1123 (23.6%) | | 77 (6.9%) |
| 11 to 13 | 1039 (21.9%) | | 113 (10.9%) |
| 14 to 16 | 658 (13.9%) | | 68 (10.3%) |
| 17 to 28 | 401 (8.4%) | | 80 (20.0%) |
| Age (years) | | | |
| <60 | 703 (14.8%) | | 110 (15.7%) |
| 60 to 69 | 1727 (36.4%) | | 174 (10.1%) |
| 70 to 79 | 1741 (36.7%) | | 155 (8.9%) |
| 80+ | 579 (12.2%) | | 55 (9.5%) |
| BMI | | | |
| Normal | 650 (14.8%) | | 42 (6.5%) |
| Overweight | 1692 (38.5%) | | 175 (10.3%) |
| Obese class I | 1236 (28.1%) | | 146 (11.8%) |
| Obese class II | 589 (13.4%) | | 76 (12.9%) |
| Obese class III | 231 (5.3%) | | 25 (10.8%) |
| Smoking | | | |
| Ex | 1776 (38.0%) | | 188 (10.6%) |
| No | 2598 (55.6%) | | 231 (8.9%) |
| Yes | 295 (6.3%) | | 71 (24.1%) |
| Drinking | | | |
| Ex | 117 (3.0%) | | 17 (14.5%) |
| No | 647 (16.5%) | | 75 (11.6%) |
| Yes | 3170 (80.6%) | | 309 (9.8%) |
| Gender | | | |
| Female | 2664 (56.1%) | | 246 (9.2%) |
| Male | 2086 (43.9%) | | 248 (11.9%) |
| IMD deprivation score (quintiles) | | | |
| 1—Least deprived | 1185 (25.0%) | | 98 (8.3%) |
| 2 | 1225 (25.8%) | | 117 (9.6%) |
| 3 | 1055 (22.2%) | | 101 (9.6%) |
| 4 | 789 (16.6%) | | 94 (11.9%) |
| 5—Most deprived | 491 (10.4%) | | 84 (17.1%) |
| Charlson Comorbidity (5-years prior) | | | |
| None | 3341 (70.3%) | | 329 (9.9%) |
| 1 | 385 (8.1%) | | 40 (10.4%) |
| 2 | 593 (12.5%) | | 63 (10.6%) |
| 3 | 189 (4.0%) | | 26 (13.8%) |
| 4+ | 242 (5.1%) | | 36 (14.9%) |
| Comorbidities | | | |
| Hypertension | 1944 (40.9%) | 279 (9.9%) | 215 (11.1%) |
| Hyperlipidaemia | 811 (17.1%) | 402 (10.2%) | 92 (11.3%) |

*(Continued)*

**Table 1.** (Continued)

| | Total | Proportion of patients with poor pain response with and without each risk factor | |
|---|---|---|---|
| | | Without | With |
| Ischaemic heart disease (IHD) | 381 (8.0%) | 446 (10.2%) | 48 (12.6%) |
| Cardiovascular disease (CVD) | 142 (3.0%) | 477 (10.4%) | 17 (12.0%) |
| Chronic obstructive pulmonary disease (COPD) | 146 (3.0%) | 470 (10.2%) | 24 (16.4%) |
| Renal failure | 625 (13.2%) | 426 (10.3%) | 68 (10.9%) |
| Cancer | 538 (11.3%) | 437 (10.4%) | 57 (10.6%) |
| Rheumatoid arthritis | 125 (2.6%) | 485 (10.5%) | 9 (7.2%) |
| Lupus | 8 (0.2%) | 493 (10.4%) | 1 (12.5%) |
| Inflammatory arthritis | 5 (0.1%) | 493 (10.4%) | 1 (20.0%) |
| Ankylosing Spondylitis | 23 (0.5%) | 492 (10.4%) | 2 (8.7%) |
| Diabetes | 573 (12.1%) | 415 (9.9%) | 79 (13.8%) |
| Knee replacement | | | |
| Total | 4212 (88.7%) | | 449 (10.7%) |
| Uni-compartmental | 538 (11.3%) | | 45 (8.4%) |
| Knee arthroscopy | 1416 (29.8%) | 288 (8.6%) | 206 (14.6%) |
| Medications | | | |
| Steroids non-GCS | 8 (0.2%) | 493 (10.4%) | 1 (12.5%) |
| Steroids GCS injections | 1829 (38.5%) | 292 (10.0%) | 202 (11.0%) |
| Steroids oral | 1136 (23.9%) | 366 (10.1%) | 128 (11.3%) |
| Prednisolone | 1119 (23.6%) | 368 (10.1%) | 126 (11.3%) |
| NSAIDs | 4226 (89.0%) | 50 (9.5%) | 444 (10.5%) |
| Opioids (full) | 2022 (42.6%) | 226 (8.3%) | 268 (13.3%) |
| Opioids (partial) | 3517 (74.0%) | 105 (8.5%) | 389 (11.1%) |
| Antibiotics | 4313 (90.8%) | 29 (6.6%) | 465 (10.8%) |
| Anticonvulsants (gabapentin, pregabalin) | 410 (8.6%) | 430 (9.9%) | 64 (15.6%) |
| Paracetamol | 3910 (82.3%) | 78 (9.3%) | 416 (10.6%) |
| Antidepressants (SSRI, TCA) | 1949 (41.0%) | 245 (8.8%) | 249 (12.8%) |
| **POST-OPERATIVE** | | | |
| Complication (3-months) | 203 (4.3%) | 463 (10.2%) | 31 (15.3%) |
| Readmission (3-months) | 568 (12.0%) | 396 (9.5%) | 98 (17.3%) |
| Re-operation (3-months) | 141 (3.0%) | 470 (10.2%) | 24 (17.0%) |
| Revision (3-months) | 8 (0.2%) | 491 (10.4%) | 3 (37.5%) |
| Manipulation under anaesthetic (3-months) | 42 (0.9%) | 481 (10.2%) | 13 (31.0%) |
| Irrigation / Debridement (3-months) | 27 (0.6%) | 488 (10.3%) | 6 (22.2%) |
| Length of stay (primary) | | | |
| < 2-days | 56 (1.2%) | | 5 (8.9%) |
| 2 to 4 days | 1808 (38.1%) | | 167 (9.2%) |
| 4 to 6 days | 1853 (39.0%) | | 200 (10.8%) |
| 6 to 10 days | 799 (16.8%) | | 87 (10.9%) |
| >10 days | 234 (4.9%) | | 35 (15.0%) |
| Number of GP consultations (3-months) | | | |
| None | 306 (6.4%) | | 24 (7.8%) |
| 1 to 4 | 2680 (56.4%) | | 235 (8.8%) |
| 5 to 9 | 1395 (29.4%) | | 185 (13.3%) |
| 10+ | 369 (7.8%) | | 50 (13.6%) |
| Medication use (3-months) | | | |

(*Continued*)

**Table 1.** (Continued)

|  | Total | Proportion of patients with poor pain response with and without each risk factor | |
|---|---|---|---|
|  |  | Without | With |
| NSAIDS | 1695 (35.7%) | 304 (10.0%) | 190 (11.2%) |
| Opioids (full) | 1681 (35.4%) | 253 (8.2%) | 241 (14.3%) |
| Opioids (partial) | 1136 (23.9%) | 368 (10.2%) | 126 (11.1%) |
| Antibiotics | 1117 (23.5%) | 340 (9.4%) | 154 (13.8%) |
| Anticonvulsants (gabapentin, pregabalin) | 191 (4.0%) | 471 (10.3%) | 23 (12.0%) |
| Paracetamol | 2241 (47.2%) | 256 (10.2%) | 238 (10.6%) |
| Antidepressants | 738 (15.5%) | 389 (9.7%) | 105 (14.2%) |

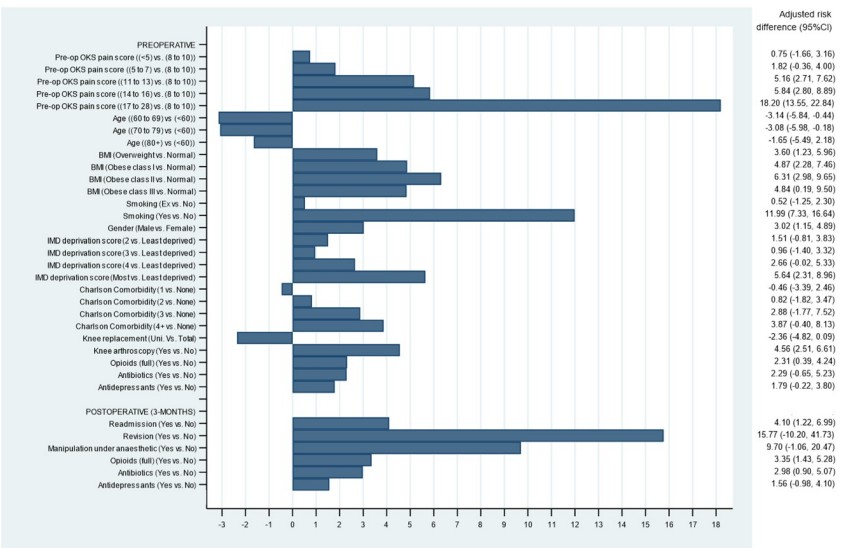

**Fig 4. Adjusted risk differences for predictors of poor pain outcomes.**

18.0%) and antibiotics (32.4% vs. 18.3%), with less marked difference for other medicines]. In terms of characteristics, those who moved to a worse pain state were more likely to be obese and have more pre-operative comorbidities (one or more comorbidities 41.2% vs. 25.95%), particularly diabetes (23.5% vs 9.8%).

**Table 2. Pain state change between pre-operative and 6-month post-operative assessments.**

| | 6-month post-op OKS pain score | | | | | | |
|---|---|---|---|---|---|---|---|
| Pre-op OKS pain score | <4 | 4 to 6 | 7 to 9 | 10 to 12 | 13 to 16 | 17 to 28 | Poor Pain outcome |
| <4 | 26 (4.6%) | 36 (6.4%) | 40 (7.1%) | 54 (9.6%) | 68 (12.1%) | 338 (60.1%) | 59 (10.5%) |
| 4 to 6 | 16 (1.7%) | 31 (3.2%) | 52 (5.4%) | 83 (8.6%) | 105 (10.9%) | 680 (70.3%) | 97 (10.0%) |
| 7 to 9 | 7 (0.6%) | 14 (1.3%) | 22 (2.0%) | 60 (5.3%) | 91 (8.1%) | 929 (82.7%) | 77 (6.9%) |
| 10 to 12 | 4 (0.4%) | 13 (1.3%) | 19 (1.8%) | 40 (3.9%) | 79 (7.6%) | 884 (85.1%) | 113 (10.9%) |
| 13 to 16 | 1 (0.2%) | 1 (0.2%) | 7 (1.1%) | 17 (2.6%) | 27 (4.1%) | 605 (92.0%) | 68 (10.3%) |
| 17 to 28 |  | 5 (1.3%) |  | 7 (1.8%) | 22 (5.5%) | 367 (91.5%) | 80 (20.0%) |

## Discussion

### Main findings

We have identified a number of risk factors that are associated with an increased risk of poor pain outcome. The strongest pre-operative risk factors were: having only mild knee pain symptoms at the time of surgery, being a current smoker, obesity, and living in the most deprived areas. Opioid and antidepressant medication use were also associated with worse pain outcomes. The strongest post-operative factors were revision surgery and manipulation under anaesthetic within three months after the operation. We identified a range of other important risk factors with more moderate effects in terms of absolute risk differences in pain outcome, including a history of previous knee arthroscopy, and use of opioids within the three months after surgery, in addition to a number of other risk factors. Those with the least pre-operative pain were more likely to move to a worse post-operative pain state and were most likely to take pain relieving medicines both pre- and post-surgery, including opioids.

### Strengths and limitations

Strengths of this study include the use of a large national linked dataset containing a wide range of clinical information from both primary and secondary care, and in the time periods both before and after surgery. The CPRD-HES linked data have previously been shown to be representative of the wider population in respect of patient demographic characteristics [15, 25]. This sample has also been compared with the mandatory National Joint Registry (NJR) in respect to knee replacement patient profile, for NJR mean age 68.9 (SD 9.6), 56.6% female. In our CPRD sample, the mean age was 69 (SD 9), of whom 56.1% were female. However, within this routine dataset, we do not have information on whether a patient received a unilateral or bilateral knee replacement and hence we are unable to exclude bilateral procedures from the dataset. Another limitation is that we are making an assumption that risk factors of poor pain outcomes, are the same for patients receiving uni-compartmental and total knee replacement. Testing for this would require test for interaction, with all other risk factors in the model, but such multiple testing could lead to type 1 errors and are in any case very low powered. Medical and surgical complications were considered as separate predictors *a priori* and others may have defined complications differently from this study. PROMs data provided a robust measure of patient-reported pain symptoms. A limitation is that national linked PROMs data has considerable levels of missing data (60.4%) in the six-month post-operative questionnaire. As the six-month data comprises our study's outcome variable, we only included patients with both pre- and six-month post-operative OKS-PS. There were very little missing data in the wide range of predictor variables included in the study, with the exception of BMI, smoking and drinking, and these variables are widely known to have missing data in CPRD. To account for potential bias, we imputed these variables using multiple imputation. A limitation is the lack of any data on coping, beliefs and expectations of outcome, and any basic mental health data, in particular of depression. The time window we used to define post-operative risk factors of up to three months after surgery was identified on the basis of agreed definitions of chronic post-surgical pain that develops or increases in intensity after a surgical procedure [26] (defined as pain 'present for at least three months' [27]), and which may provide a useful window in which clinicians could identify patients and intervene to prevent the development of persistent and potentially intractable pain.

### What is already known

Patients with worse pre-operative pain achieve greater change in symptoms (journey), whilst those with mild pre-operative pain have less change but retain the greatest post-operative level

of pain and mildest symptoms (destination) [10, 28]. Pre-operative factors age [29], gender [30], obesity [31], social deprivation [7], co-morbidity [32] and smoking [33] are known to influence the surgical outcome, as does having a uni-compartmental knee replacement [34] and the influence of previous non-replacement knee surgery such as knee arthroscopy [35]. This study's unique contribution is our focus on knee pain as the outcome, and our work to identify patients who do not respond to surgery expressed in terms of adjusted absolute risk. Some other studies have looked at composite outcomes combining symptoms of pain, stiffness and function. Predictors of pain outcomes are not necessarily the same as functional outcomes, and likewise a patient may have improvement in pain symptoms, but not in function [7].

Fewer studies have tried to identify post-operative risk factors [36], and a strength of our study is the focus on primary care risk factors and medication use that is not usually available in other routine datasets. The operative predictors of revision surgery and manipulation under anaesthetic are to be expected, as these indicate that surgery has failed and are indicative of poor outcome. Risk factors such as revision surgery, medical complications and re-operations are uncommon, but should serve as flags to indicate these patients are at risk of chronic pain and may need to be seen by a specialist with knowledge of pain prevention and management. Patients using opioid medication prior to knee replacement are at increased risk of post-operative complications including opioid overdose [37, 38] and have been the subject of increasing attention [39]. Opioids are commonly used in the management of pain both before and after knee replacement despite a lack of recommendation for the use of any types of opioids other than Tramadol, and the evidence of complications [40, 41]. Monitoring analgesic use such as opioids post-operatively can be an indicator of persistent pain following arthroplasty [42] in addition to indicating a poor pain outcome [43]. Use of antidepressants is an important factor to explore, because pain and depression are known to be associated and depression may be a key area for intervention that may help to improve pain coping and management [44]. Antidepressants are commonly used in patients with refractory pain or who have a neuropathic component to their pain [43, 45]. Although pre-operative psychological distress is associated with chronic pain, specific psychological risk factors for chronic pain after knee replacement have been identified with possible management strategies indicated [46].

## What this study adds

It is still unclear why patients with little knee pain are undergoing knee replacement and there is a need to understand how decisions about such patients are made. Risk factors of smoking and obesity already feature within NICE guidelines [1], but although weight loss is recommended, the guidelines are clear that 'patient-specific factors (including smoking, obesity and comorbidities) should not be barriers to referral for joint surgery'. Whilst the majority of patients in these groups do achieve good pain outcomes and should not be denied access to care, they are at an increased absolute risk of poor pain outcome. In the three-month window after surgery, there is an opportunity to provide interventions for patients that reduce the risk of poor pain outcome. Opioid medication prescribing represents an important target for future interventions. Joint replacement is effective but there is a need to focus on careful patient selection and modifiable risk factors to reduce poor outcomes. For those whose outcomes can be optimised it is timely to explore interventions that target such risk factors both before and after surgery.

## Supporting information

**S1 Table. Imputation model checks with descriptive statistics and univariable model logistic regression results comparing complete case to imputed data.**
(DOCX)

**S1 Fig. Receiver operating characteristic curve (ROC) curve for discriminatory ability of variables in the final model.**
(DOCX)

## Acknowledgments

This study is based in part on data from the CPRD obtained under licence from the UK Medicines and Healthcare products Regulatory Agency. However, the interpretation and conclusions contained in this study are those of the authors alone.

## Author Contributions

**Writing – review & editing:** Hasan Raza Mohammad, Rachael Gooberman-Hill, Antonella Delmestri, John Broomfield, Rita Patel, Joerg Huber, Cesar Garriga, Christopher Eccleston, Rafael Pinedo-Villanueva, Tamer T. Malak, Nigel Arden, Andrew Price, Vikki Wylde, Tim J. Peters, Ashley W. Blom, Andrew Judge.

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
