## [Decision Letter · Decision Letter 0]

14 Oct 2021

PONE-D-21-27576Risk factors associated with poor pain outcomes following primary knee replacement surgery:analysis of data from the Clinical Practice Research Datalink, Hospital Episode Statistics and Patient Reported Outcomes as part of the STAR research programmePLOS ONE

Dear Dr. Judge,

Thank you for submitting your manuscript to PLOS ONE. After careful consideration, we feel that it has merit but does not fully meet PLOS ONE’s publication criteria as it currently stands. Therefore, we invite you to submit a revised version of the manuscript that addresses the points raised during the review process.

We look forward to receiving your revised manuscript.

Kind regards,

Armando Almeida

Academic Editor

PLOS ONE

Journal Requirements:

When you resubmit, please ensure that you provide the correct grant numbers and funders name for the awards you received for your study in the ‘Funding Information’ section.

“I have read the journal's policy and the authors of this manuscript have the following competing interests:

Competing interest statement

All authors have completed the Unified Competing Interest form at www.icmje.org/coi_disclosure.pdf and declare: AJ reports grants from NIHR PGfAR, during the conduct of the study; personal fees from Freshfields Bruckhaus Derringer, personal fees from Anthera Pharmaceuticals Ltd, outside the submitted work; RGH reports grants from NIHR PGfAR, during the conduct of the study. HRM reports grants from Royal College of Surgeons Research Fellowship, outside the submitted work; RPV reports research funding from UK-NIHR, Kyowa Kirin Services, International Osteoporosis Foundation, and lecture fees and/or consulting honoraria from Amgen, UCB, Kyowa Kirin Services, and Mereo Biopharma, all outside of the scope of this study; NA reports personal fees from Pfizer/Lilly, personal fees from Bristows LLP, grants from Merck Grant,  outside the submitted work; VW reports grants from NIHR, during the conduct of the study; TJP reports grants from UK NIHR Programme Grant for Applied Research, during the conduct of the study. All other authors declare no conflicts of interest.”

Reviewers' comments:

Reviewer's Responses to Questions

**Comments to the Author**

1. Is the manuscript technically sound, and do the data support the conclusions?

Reviewer #1: Partly

Reviewer #2: Yes

Reviewer #3: Yes

2. Has the statistical analysis been performed appropriately and rigorously? 

Reviewer #1: No

Reviewer #2: Yes

Reviewer #3: I Don't Know

3. Have the authors made all data underlying the findings in their manuscript fully available?

Reviewer #1: Yes

Reviewer #2: No

Reviewer #3: No

4. Is the manuscript presented in an intelligible fashion and written in standard English?

Reviewer #1: Yes

Reviewer #2: Yes

Reviewer #3: Yes

5. Review Comments to the Author

Reviewer #1: The authors present a study with a large sample size focusing on risk factors for poor pain outcomes after joint replacement. This is a relevant field of research, but it is my opinion that this manuscript would benefit from a clearer account of the state of the art, to show the readership why this is a novel and relevant study. Additionally, the presentation of results could also be more informative and complete. More specific comments are detailed below.

INTRODUCTION

- Globally, it is my opinion that the introduction section could be more complete, to provide an overview of the state-of-the-art in this field and to show the readership why the present study is warranted. What risk factors for poor outcomes have been analyzed in previous studies? Have the results been consistent across studies? Which risk factors are the most important? Which ones should be further investigated? It is not clear what the present study might add as novelty to the field.

- 1st paragraph, line 1: Should be “have”

- The authors state that “Identifying potentially modifiable risk factors might enable innovation and interventions to reduce the numbers of future patients who experience pain after surgery.” However, few of the analyzed risk factors are modifiable. How does this statement relate to the rest of the work?

METHODS

- It would be more accurate to use the subtitle “Sample” instead of “Population”.

- The variable IMD should be explained since this acronym is not known across all countries.

- In the same line, the Charlson Comorbidity Index should also be briefly explained.

- Why were these specific variables included in the model? There is no information concerning multicollinearity.

Statistical analysis

- Line 1: Patients’

- What is the rationale to categorize the OKS score into 6 categories?

- What is the rationale to exclude underweight BMI?

RESULTS

*Page 6, Pre-operative predictors

- Table 1: Why was the column “No” only filled in for some variables? Using age as an example, it would be possible to report the number of patients with poor outcomes who were “not” in the <60 group. Presumably, this would be the sum of patients who had a poor outcome and were in the other age groups.

- What was the rationale to choose the reference category for each variable? For example, Pre-op OKS score groups are all only compared with the 8-10 score, age groups are all just compared against the <60 group…

- For age, it is stated that “those aged between 60-79 had a reduced risk of poor pain outcome compared with the oldest and youngest age groups”. However, there is no 60-79 group and these age groups (60-69 and 70-79 are not compared with older age groups (only with <60). Can the authors please clarify what is meant?

- I suggest rewriting the sentence “Poor pain outcomes were more likely in current smokers, males, people living in the most deprived areas and those with inflammatory arthritis”. As is, it seems to the reader that the authors are establishing these predictors as the most important ones. Also, I do not believe that these conclusions can be totally inferred from Table 1. I would suggest completing this table with additional statistical information for each predictor.

- I could not find results concerning inflammatory arthritis in the tables or graphs.

- For the remaining predictors, please note that, according to figure 3, effect is are non-significant for Charlson (95% CI=0.99, 1.90), Uni vs. Total arthroplasty (95% CI=0.58, 1.04) and Antidepressants (95% CI=0.98, 1.45) (the CIs include 1).

*Page 7, Absolute risk differences:

- Figure 4: This graph would be more informative if the ARD and CI values were detailed for each predictor, so that the reader can judge the statistical significance of each one.

- What criteria were used to determine what is a “large” ARD?

- When the CI are presented in brackets, the information that the values are the 95% CI should be present, and not only the numbers (e.g., 95% CI: 7.3,16.6)

- Did the authors consider presenting the ROC curve for the model? I believe this would provide a more visual idea of the model.

Page 8, Pain state change:

- “most likely to not improve” – It seems that Table 2 only provides descriptive information. It is not clear how the authors ascertained which patients were most likely to improve or which statistical inference test was used to reach this conclusion.

- “we characterized the pre-operative and post-operative characteristics of those with worsening pain state change.” – Did the authors conduct any statistical test to compare the groups? These conclusions cannot be inferred from descriptive statistics alone. The complete results and p values should be detailed to the reader.

- “There were large differences in” – Was there a formal effect size calculation for between-group differences?

DISCUSSION

- “it may be worthwhile providing pre-surgical interventions to address modifiable risk factors (smoking, obesity and comorbidities)”. According to the results, there is a reduced risk of poor outcomes for yes vs. non smokers, but not for ex vs. non smokers. Therefore, this study does not seem to support pre-surgical interventions to address smoking. Also, the analysis of comorbidities did not show significant results (all CI include 1). Though the identification of risk factors is important, this study mainly focuses on non-modifiable characterizes and thus the clinical implication should be reconsidered.

- Most other studies have looked at composite outcomes combining symptoms of pain, stiffness and function.” – I believe that this may be an overstatement, since many studies in this field analyze pain as an outcome. Please reconsider if this is accurate information.

Reviewer #2: This manuscript reports the results of a study that uses an existing dataset to test for predictors for poor pain outcomes following total knee replacement (TKR). Such studies are common in the field of TKR, but with varying predictors, outcomes and methods. This study is a useful addition to the current body of evidence although, for a study that uses an existing dataset and fairly routine statistical methods, it is surprising to see 16 authors listed. Specific comments follow but these are minor comments and I have no major problems with the paper which is well written, well presented and well conducted.

1. I think the abstract should contain the time point of the outcome

2. The study involves data linkage. This is a process commonly associated with surprisingly high error rates. Some data on the matching rates and processes would be useful.

3. What was done about bilateral procedures – they should either have been excluded from the dataset or included in the model?

4. Population selection: can any information be provided on the likely representativeness of this sample to the population as a whole? I know this is addressed in the discussion but this is based on previous analyses, not this exact dataset.

5. The study reports relative risk ratios (instead of the usual odds ratios generated by logistic regression). Just checking that this is correct – that the authors converted the ORs to RRs?

6. Using the Oxford PS (pain score) as a predictor seems odd when this is the score used to define (calculate) the outcome. I realise that the Oxford PS score and the treatment effect (based on that score) are different but surely those with a worse pre-operative pain score will tend to have larger treatment effects because they have more “room” to improve? This is why I don’t like using treatment effect – I would rather know how much pain they have at 6 months. Pre-operative pain can be added to the model to adjust for its effect. But I am open to arguments to the contrary.

7. The pre-op associations were consistent with other studies and not surprising (except the association with prior knee arthroscopy, which was interesting, and gender was the opposite of what I have seen before). I have a problem with using post-operative data to predict early post-operative outcomes. For example, those taking opioids post-operatively were more likely to have poor pain outcomes. These two variables are kind of measuring the same thing: post-op pain, and it is unlikely that ceasing these drugs will prevent pain at 6 months. Similarly, those who needed further surgery and had complications were more likely to have pain. These findings are expected and don’t point to any clinically useful knowledge, apart from avoiding complications. I am not asking the authors to remove them, but some comment in the discussion about the limited usefulness and obviousness of these findings should be mentioned.

Ian Harris

Professor of Orthopaedic Surgery, UNSW Sydney

Reviewer #3: The manuscript reports the findings from an observational study aiming to identify risk factors for poor pain outcomes after TKR. Pain outcomes were defined using the pain section of the OKS.

ABSTRACT

In the manuscript, complications seem important, but are not mentioned here?

INTRODUCTION

1. Suggest modify sentence – ‘but around 1 in 5 will continue to experience pain…’ The 20% isn’t upheld in many studies. Suggest change to “the % experience ongoing pain is variable (add refs) with up to 20% experiencing…”. The current data also supports the finding that 20% is not often upheld.

(Suggest the authors could do a systematic review on whether the % with persistent pain has improved across time? (as another study) as this may explain why 20% seems an outdated value now)

METHODS

1. The introduction talks about TKA for people with OA. Clarify if only people with OA are in the dataset used. If not, change the Intro to be more inclusive of other indications for surgery.

2. Justify inclusion of unicompartmental surgery. Its inclusion implies the same predictors will apply

3. Please elaborate on justification for included complications. Were these defined by stakeholders? Were they chosen on severity? This is important as you include UTI (minor) to most severe and transparency is required here.

4. Clarify total number of GP visits? From surgery to 3 months post-surgery

5. Mention model fit statistics and performance test to be used here (mentioned in Results)

6. Clarify medication use includes pre and post-op. It gets confusing when talking about med use in Results.

RESULTS

1. Missing data for alcohol consumption of 17% is high. Should do sensitivity analysis without that variable otherwise justify why that is not necessary.

2. Clarify you checked for correlations between medication use pre and post-op. Same with opioid use and pain pre and post-op? There may be collinearity there.

Table 1

Clarify that ‘complication’ is different to say surgery for MUA? I would argue MUA is a subset of complication (as this applies to some other complications too). By keeping them separate, this assumes different complications have different ‘weights’ so to speak. Can you justify/explain this approach. Seems like you have distinguished ‘medical’ from surgical and surgical is broken down further?

Fig 1 - Please justify exclusion of underweight BMI

DISCUSSION

1. Discussing use of opioids post-operatively as a risk factor along side pre-op factors is confusing. It makes sense that opioid use is assoc with persistent pain if the pain is driving use. On the other hand, pre-op BMI as a predictor is completely different. It may be a predictor as opposed to opioid use which may not ‘predict’, but rather be reactionary.

Can the authors try to tease this out or dela with this better.

6. PLOS authors have the option to publish the peer review history of their article (what does this mean?). If published, this will include your full peer review and any attached files.

Reviewer #1: No

Reviewer #2: **Yes: **Ian A Harris

Reviewer #3: No

---

## [Author Response · Author response to Decision Letter 0]

17 Nov 2021

Dear Dr Almeida,

Thank you for reviewing our manuscript at PLOS ONE as per your email date 15 Oct 2021. We have considered these comments carefully, and our responses and amendments to the manuscript are reported below.

1. Reviewer #1: The authors present a study with a large sample size focusing on risk factors for poor pain outcomes after joint replacement. This is a relevant field of research, but it is my opinion that this manuscript would benefit from a clearer account of the state of the art, to show the readership why this is a novel and relevant study. Additionally, the presentation of results could also be more informative and complete. More specific comments are detailed below. 

Thank you for your comments. We have revised the manuscript to give a clearer account of current knowledge, describe our study and improved the presentation of results. 

[see Introduction and Results, and below]

2. INTRODUCTION - Globally, it is my opinion that the introduction section could be more complete, to provide an overview of the state-of-the-art in this field and to show the readership why the present study is warranted. What risk factors for poor outcomes have been analyzed in previous studies? Have the results been consistent across studies? Which risk factors are the most important? Which ones should be further investigated? It is not clear what the present study might add as novelty to the field. 

Most previous studies have focused on pre-operative risk factors when assessing predictors of patients reported outcomes. Many studies have looked at composite outcomes that combine information on pain, stiffness and function, whereas patients tell us that the most important outcome to them is pain. Our previous research found that the pre-operative risk factors were different for pain and functional domains. In particular, the discriminatory ability of the models to predict function were much better, than for pain outcome, where the discriminatory ability of pre-operative risks factors was poor (https://pubmed.ncbi.nlm.nih.gov/22532699/). Within the literature less is known about whether post-operative risk factors can help improve prediction. 

The specific aim of our study was to use a wide and comprehensive range of information on predictors, that covered both pre and post-operative information, to see if this could help improve our prediction and better identify patients at risk of post-surgical chronic pain.

We have now expanded the introduction to provide a more complete overview of the rationale for this study.

[Introduction, paragraph 2]

3. - 1st paragraph, line 1: Should be “have” 

Amended [Introduction, paragraph 1]

4. - The authors state that “Identifying potentially modifiable risk factors might enable innovation and interventions to reduce the numbers of future patients who experience pain after surgery.” However, few of the analyzed risk factors are modifiable. How does this statement relate to the rest of the work? 

We agree with the comment and have removed this sentence.

[Introduction, paragraph 2]

5. METHODS

 - It would be more accurate to use the subtitle “Sample” instead of “Population”. 

Amended [Methods, paragraph 3]

6. - The variable IMD should be explained since this acronym is not known across all countries. 

We agree and have added the following sentences:

To measure socioeconomic deprivation, we used the index of multiple deprivation (IMD) score. This is a relative measure of deprivation for small areas – termed lower super output areas (LSOAs) – which are defined as geographical areas of a similar population size, with an average of 1,500 residents [19]. The IMD comprises seven measures of deprivation: income deprivation; employment deprivation; education, skills and training deprivation; health deprivation and disability; crime; barriers to housing and services; and living environment deprivation. We used the IMD rank for a patient’s LSOA and categorised patients into quintiles based upon the national ranking of local areas, with quintile 1 being the least deprived group and quintile 5 being the most deprived group (i.e. reordered to aid reporting). 

[Methods, Pre-operative predictors]

7. - In the same line, the Charlson Comorbidity Index should also be briefly explained. 

We agree and have added the following sentences:

As a measure of comorbidity we used the Royal College of Surgeons’ (RCS) Charlson Score, which is calculated based on the presence of several chronic conditions identified using ICD-10 codes at the time of knee replacement surgery admission and all admissions in the preceding 5 years [20].

[Methods, Pre-operative predictors]

8. - Why were these specific variables included in the model? There is no information concerning multicollinearity. 

In discussion with clinicians we sought to identify variables within this routinely collected dataset which may provide a wide range of potential risk factors/predictors for the pain outcome. We have assessed for evidence of collinearity using variance inflation factors and there was no evidence of multicollinearity. We have added statements regarding both points. 

Predictor variables: In discussion with clinicians we sought to identify variables within this routinely collected dataset which may provide a wide range of potential predictors for the pain outcome.

We have assessed for evidence of collinearity using variance inflation factors and there was no evidence of multicollinearity.

[Methods, Predictor variables and paragraph 11]

9. Statistical analysis – Line 1: Patients’ 

Amended [Statistical analysis, paragraph 1]

10. - What is the rationale to categorize the OKS score into 6 categories? 

One of the assumptions of a regression model is linearity of continuous variables with the outcome – we used fractional polynomial regression models to assess this and there was strong evidence of non-linearity of pre-op knee scores, so to satisfy the assumption we categorised the OKS score.

11. - What is the rationale to exclude underweight BMI? 

We have added a sentence to clarify this point: 

We excluded nine underweight BMI patients, as there were too few patients in this category, leading to a small cell problem in the multivariable regression model, and it would be inappropriate to combine underweight and normal BMI categories together.

[Statistical analysis, paragraph 2]

12. RESULTS Page 6, Pre-operative predictors – Table 1: Why was the column “No” only filled in for some variables? Using age as an example, it would be possible to report the number of patients with poor outcomes who were “not” in the <60 group. Presumably, this would be the sum of patients who had a poor outcome and were in the other age groups. 

To clarify Table 1 column headings No and Yes refer to whether the risk factor is present or not, rather than whether pain is present or not. For example, for the hypertension row the No column refers to those with no hypertension who have pain. The Yes column those with hypertension and pain. Therefore, for those aged <60 years with pain (Yes column=15.7%), the No column would have to contain all those that are >=60 years with pain. We could remove the No column and add two rows per binary outcome (example below). However, for now we have left the table unchanged. We will revise the table should you feel strongly that it should be changed. 

13. - What was the rationale to choose the reference category for each variable? For example, Pre-op OKS score groups are all only compared with the 8-10 score, age groups are all just compared against the <60 group… 

Our rational for choosing the reference category is the usual practice of selecting the group with the largest sample size, as our dataset is fairly large, we selected the lowest category for each variable, we are happy to change reference categories if you feel strongly regarding this point.

14. - For age, it is stated that “those aged between 60-79 had a reduced risk of poor pain outcome compared with the oldest and youngest age groups”. However, there is no 60-79 group and these age groups (60-69 and 70-79 are not compared with older age groups (only with <60). Can the authors please clarify what is meant? The effect size is bigger in the middle age groups 

We have amended this sentence to clarify its meaning:

A ‘U-shaped’ effect was observed for patient age at surgery, where those aged between 60-69 and 70-79 had a reduced risk of poor pain outcome compared with the youngest age group. The oldest age group was similar to the youngest.

[Results, paragraph 3]

15. - I suggest rewriting the sentence “Poor pain outcomes were more likely in current smokers, males, people living in the most deprived areas and those with inflammatory arthritis”. As is, it seems to the reader that the authors are establishing these predictors as the most important ones. Also, I do not believe that these conclusions can be totally inferred from Table 1. I would suggest completing this table with additional statistical information for each predictor. 

At this stage of the results we are examining the descriptive statistics, looking at associations in the data and we are highlighting those that appear the most clinically relevant. There is no formal hypothesis testing being done, this is done with the regression modelling. Please see earlier response about missing data in Table 1. We have rewritten the sentence: The descriptive associations suggest poor pain outcomes were more likely in current smokers, males, people living in the most deprived areas and those with inflammatory arthritis (Table 1).

[Results, paragraph 3]

16. - I could not find results concerning inflammatory arthritis in the tables or graphs. 

Inflammatory arthritis appears in Table 1 below the subheading ‘Comorbidities’. No individual comorbidities appear in the forest plot as they are rare, some of the categories are too small so we present findings for combined comorbidities in the Charleston score. We have added a sentence to the methods to clarify this: 

Composite variables were used if individual characteristics were rare.

[Statistical analysis, paragraph 2]

17. - For the remaining predictors, please note that, according to figure 3, effect is are non-significant for Charlson (95% CI=0.99, 1.90), Uni vs. Total arthroplasty (95% CI=0.58, 1.04) and Antidepressants (95% CI=0.98, 1.45) (the Cis include 1). 

Please see paper by Sterne et al BMJ (https://www.bmj.com/content/322/7280/226.1.full). We are examining the strengths of association and not arbitrary measures of statistical significance with cut offs of for example p<0.05 or the related concept of whether the confidence interval includes the null value. To clarify we have added this sentence:

For the following variables there is weak evidence of an association.

[Results, paragraph 3]

18. *Page 7, Absolute risk differences:

- Figure 4: This graph would be more informative if the ARD and CI values were detailed for each predictor, so that the reader can judge the statistical significance of each one. 

Thank you we agree, and have added the absolute risk difference and confidence intervals to Figure 4.

19. - What criteria were used to determine what is a “large” ARD? 

We did not use any specific criteria, rather this was in respect of ‘strength of association’ as per the Bradford Hill criteria. We have now amended the wording: The other pre-operative variables with some absolute adjusted risk differences were… 

[Results, Absolute risk differences]

20. - When the CI are presented in brackets, the information that the values are the 95% CI should be present, and not only the numbers (e.g., 95% CI: 7.3,16.6) 

Amended [Abstract and Results, paragraph 6]

21. - Did the authors consider presenting the ROC curve for the model? I believe this would provide a more visual idea of the model. 

Thank you we agree and have now added the ROC curve to the Supplementary data file and Results, paragraph 7. 

22. Page 8, Pain state change:

- “we characterized the pre-operative and post-operative characteristics of those with worsening pain state change.” – Did the authors conduct any statistical test to compare the groups? These conclusions cannot be inferred from descriptive statistics alone. The complete results and p values should be detailed to the reader.

This is descriptive not hypothesis testing so we have amended the language and made it clear in the methods and results.

To describe their change in pain state before and 6 months after surgery, patients are categorised into pain groups, and descriptive statistics (without statistical testing) used to describe the number (percentage) of patients that move between different pain states before and after surgery.

… we describe the pre-operative and post-operative characteristics of those with worsening pain state change.

[Statistical analysis, paragraph 1 and Results, Pain state change] 

23. - “There were large differences in” – Was there a formal effect size calculation for between-group differences?

We agree and have amended wording: 

There were some differences in length of stay, where those who moved to a worse pain state had much shorter length of stay

[Results, Pain state change]

24. DISCUSSION – “it may be worthwhile providing pre-surgical interventions to address modifiable risk factors (smoking, obesity and comorbidities)”. According to the results, there is a reduced risk of poor outcomes for yes vs. non smokers, but not for ex vs. non smokers. Therefore, this study does not seem to support pre-surgical interventions to address smoking. Also, the analysis of comorbidities did not show significant results (all CI include 1). Though the identification of risk factors is important, this study mainly focuses on non-modifiable characterizes and thus the clinical implication should be reconsidered. 

We agree and have removed the statement. 

[Discussion, paragraph 5-What this study adds]

25. - Most other studies have looked at composite outcomes combining symptoms of pain, stiffness and function.” – I believe that this may be an overstatement, since many studies in this field analyze pain as an outcome. Please reconsider if this is accurate information. 

Scores like the Western Ontario and McMaster Universities Osteoarthritis Index (WOMAC) and Oxford Knee Score are commonly used composite outcomes. We have amended wording to clarify our meaning. 

Some other studies have looked at composite outcomes

[Discussion, paragraph 3-What is already known]

26. Reviewer #2: This manuscript reports the results of a study that uses an existing dataset to test for predictors for poor pain outcomes following total knee replacement (TKR). Such studies are common in the field of TKR, but with varying predictors, outcomes and methods. This study is a useful addition to the current body of evidence although, for a study that uses an existing dataset and fairly routine statistical methods, it is surprising to see 16 authors listed. Specific comments follow but these are minor comments and I have no major problems with the paper which is well written, well presented and well conducted. 

27. 1. I think the abstract should contain the time point of the outcome 

Amended the objective to: Identify risk factors for poor pain outcomes six months after primary knee replacement surgery.

[Abstract]

28. 2. The study involves data linkage. This is a process commonly associated with surprisingly high error rates. Some data on the matching rates and processes would be useful. 

The data linkage is done in-house by CPRD and not by the researchers themselves. Providers of data such as CPRD and NHS Digital provide the service for data linkage as a ‘trusted third party’ in a secure environment. This ensures the researchers themselves only have access to the anonymised linked data, and for information governance reasons, there is no need for sharing of personal data. Information on CPRD linkage is available here https://cprd.com/linked-data. 

Hence researchers who use data provided by CPRD do not have information on the data linkage process itself. 

We now describe in the methods of the paper that: Linkage of CPRD-HES-ONS-PROMS data is done by NHS Digital as a ‘trusted third party’.

[Methods, paragraph 2]

29. 3. What was done about bilateral procedures – they should either have been excluded from the dataset or included in the model? 

Within this routine dataset, we do not have information on whether or not a patient received a unilateral or bilateral knee replacement. This is now described as a limitation in the discussion section.

However, within this routine dataset, we do not have information on whether a patient received a unilateral or bilateral knee replacement and hence we are unable to exclude bilateral procedures from the dataset. 

[Discussion, Strengths and limitations]

30. 4. Population selection: can any information be provided on the likely representativeness of this sample to the population as a whole? I know this is addressed in the discussion but this is based on previous analyses, not this exact dataset. 

Data from the national joint registry provides information on patients receiving hip and knee replacement where there is mandatory data collection nationally. To address the reviewers question, we have compared the mean age, and sex of patients in our CPRD sample, to that of the NJR https://reports.njrcentre.org.uk/Portals/0/PDFdownloads/NJR%2018th%20Annual%20Report%202021.pdf. For knee replacement in NJR mean age 68.9 (SD 9.6), 56.6% female. In our CPRD sample mean age of 69 (SD 9) of whom 56.1% were female. Hence very comparable in respect of age and gender. This has been added in the discussion section.

This sample has also been compared with the mandatory National Joint Registry (NJR) in respect to knee replacement patient profile, for NJR mean age 68.9 (SD 9.6), 56.6% female. In our CPRD sample, the mean age was 69 (SD 9), of whom 56.1% were female. 

[Discussion, Strengths and limitations]

31. 5. The study reports relative risk ratios (instead of the usual odds ratios generated by logistic regression). Just checking that this is correct – that the authors converted the Ors to RRs? 

We confirm that in this study we are indeed using relative risk ratios. Logistic regression is appropriate to use for relatively infrequent outcomes, whereas for more common outcomes, odds ratios are not a good approximation for relative risk. Hence in this paper, we have presented the results as relative risk ratios by fitting a generalized linear model with a binomial error structure and a log link function (log-logistic model) in order to estimate the relative risk.

32. 6. Using the Oxford PS (pain score) as a predictor seems odd when this is the score used to define (calculate) the outcome. I realise that the Oxford PS score and the treatment effect (based on that score) are different but surely those with a worse pre-operative pain score will tend to have larger treatment effects because they have more “room” to improve? This is why I don’t like using treatment effect – I would rather know how much pain they have at 6 months. Pre-operative pain can be added to the model to adjust for its effect. But I am open to arguments to the contrary. 

We understand the point that the reviewer is making, and there is some debate in the literature as to how to define patient reported outcomes in observational studies with repeated measures repeated before and after surgery. This is discussed for example, in the article by Losina and Katz, as to whether interest is in the ‘journey’ (that is, absolute or relative change in pain score before and after surgery), or the ‘destination’ (that is, the attained post-operative level of pain). https://www.ncbi.nlm.nih.gov/pmc/articles/PMC3448885/

The limitation with a continuous OKS pain score outcome adjusting for baseline OKS as a covariate (such as in analysis of covariance ANCOVA models) is that, for knee replacement, the majority of patients achieve substantial improvements in pain, and we end up identifying predictors of patients with excellent improvement in pain, compared with patients receiving really good improvements of pain https://online.boneandjoint.org.uk/doi/pdf/10.1302/0301-620X.94B3.27425. Whereas our actual interest is in trying to identify the minority of patients at the tail of the distribution who do not achieve good improvement in symptoms, and hence to define a binary outcome according to whether or not patients had a good pain outcome after surgery. Our preference is to use the treatment effect to define outcome, and we have previously reported on this approach and validated it (Huber J, Husler J, Dieppe P, et al. A new responder criterion (relative effect per patient (REPP) > 0.2) externally validated in a large total hip replacement multicenter cohort (EUROHIP). Osteoarthritis and Cartilage 2016;24(3):480-3.)

The rationale for our descriptive analyses, looking at pain state change, was to better understand the comment made by the reviewer, that patients with larger treatment effects have more room to improve. We noted that patients with the mildest pre-operative pain symptoms were most likely to not improve and their pain actually got worse. Specifically, they moved to a worse pain state following surgery, where 20% of patients had a poor pain outcome compared with around 10% in the other pre-operative pain states. We characterised why this might be, where these patients with mild pre-operative symptoms who experienced worse outcomes had more complications and re-operations. 

33. 7. The pre-op associations were consistent with other studies and not surprising (except the association with prior knee arthroscopy, which was interesting, and gender was the opposite of what I have seen before). I have a problem with using post-operative data to predict early post-operative outcomes. For example, those taking opioids post-operatively were more likely to have poor pain outcomes. These two variables are kind of measuring the same thing: post-op pain, and it is unlikely that ceasing these drugs will prevent pain at 6 months. Similarly, those who needed further surgery and had complications were more likely to have pain. These findings are expected and don’t point to any clinically useful knowledge, apart from avoiding complications. I am not asking the authors to remove them, but some comment in the discussion about the limited usefulness and obviousness of these findings should be mentioned. 

This study has been conducted as part of a wider NIHR programme grant on chronic knee pain, where the a priori aim was to see if identification and inclusion of post-operative risk factors, could help us to identify patients who are most at risk of chronic post operative pain, rather than using pre-operative risk factors alone. The funding application for the NIHR programme grant went through a rigorous process of peer review, and this is the research question we have been funded to address. 

The reviewers’ points relating to opioid use, early complications, and further surgery, are well made, and we have included this within the discussion section. However, knowledge of such post-operative factors would help us to identify patients earlier, even if by a few months, allowing us to intervene earlier and ensure such patients are receiving appropriate pain management services.

34. Ian Harris Professor of Orthopaedic Surgery, UNSW Sydney 

Reviewer #3: The manuscript reports the findings from an observational study aiming to identify risk factors for poor pain outcomes after TKR. Pain outcomes were defined using the pain section of the OKS. 

35. ABSTRACT

In the manuscript, complications seem important, but are not mentioned here? 

These have now been included in the abstract.

Those patients with worsening pain state change had more complications by 3-months (11.8% among those in a worse pain state vs. 2.7% with the same pain state).

36. INTRODUCTION

1. Suggest modify sentence – ‘but around 1 in 5 will continue to experience pain…’ The 20% isn’t upheld in many studies. Suggest change to “the % experience ongoing pain is variable (add refs) with up to 20% experiencing…”. The current data also supports the finding that 20% is not often upheld. 

The figure of 20% was determined through a systematic review of the literature by Beswick et al (https://pubmed.ncbi.nlm.nih.gov/22357571/). We acknowledge the reviewer’s point that this was published some time ago now, and that this estimate will vary between studies and change over time. Hence, we have modified this statement as suggested.

Many patients can expect to achieve reductions in knee pain and improvements in functional outcomes [3]. The percentage who experience ongoing pain is variable [4], with up to 20% experiencing knee pain that impacts their quality of life [5].

[Introduction, paragraph 1]

37. (Suggest the authors could do a systematic review on whether the % with persistent pain has improved across time? (as another study) as this may explain why 20% seems an outdated value now) 

We agree with the reviewer that this will be a useful area for a future study, and have mentioned this in the discussion section, but is beyond the scope of this particular paper.

38. METHODS 

1. The introduction talks about TKA for people with OA. Clarify if only people with OA are in the dataset used. If not, change the Intro to be more inclusive of other indications for surgery. 

The reviewer is correct that this paper contains patients receiving a primary elective knee replacement for all indications, and not just osteoarthritis. We have changed the introduction to be more inclusive of other indications.

Over 100,000 knee replacement operations are carried out each year in the UK for osteoarthritis and other surgical indications [2]

[Introduction, paragraph 1]

39. 2. Justify inclusion of unicompartmental surgery. Its inclusion implies the same predictors will apply We included all patients receiving knee replacement surgery, whether total or uni-compartmental. 

We included total versus uni-compartmental as a predictor in the model, to see if this was associated with worse pain outcomes, with weak evidence of unicompartmental patients having lower risk of worse pain outcomes. 

A limitation of this, as the reviewer highlights, is that we are making an assumption - namely, that risks factors of poor pain outcomes, are the same for patients receiving uni-compartmental and total knee replacement. Investigating this would require tests for interactions, with all other risk factors in the model, but such multiple testing could lead to type 1 errors being made, and are in any case very low powered.

We have therefore made it clear in the methods that the population comprises patients receiving primary total and uni-compartmental knee replacement [Methods, Sample paragraph 3]. In the discussion, we describe this limitation, and the assumption being made for other risk factors identified.

Another limitation is that we are making an assumption that risk factors of poor pain outcomes, are the same for patients receiving uni-compartmental and total knee replacement. Testing for this would require test for interaction, with all other risk factors in the model, but such multiple testing could lead to type 1 errors and are in any case very low powered. 

[Discussion, Strengths and limitations]

40. 3. Please elaborate on justification for included complications. Were these defined by stakeholders? Were they chosen on severity? This is important as you include UTI (minor) to most severe and transparency is required here. 

Complications were chosen a-priori in discussion with a number of orthopaedic surgeons, as to what were considered to be clinically relevant complications of knee replacement surgery. The chosen complications have been described and published previously https://www.oarsijournal.com/article/S1063-4584(14)01051-6/fulltext

We have added this reference to the Methods, Post-operative predictors.

41. 4. Clarify total number of GP visits? From surgery to 3 months post-surgery 

Amended 

and calculated the total number of general practice visits between surgery and 3 months post-surgery 

[Methods, paragraph 11]

42. 5. Mention model fit statistics and performance test to be used here (mentioned in Results) 

Added to Statistical analysis: 

The C-statistic was used to describe the discriminatory ability of variables in the final model.

43. 6. Clarify medication use includes pre and post-op. It gets confusing when talking about med use in Results. 

Amended 

We identified medications prescribed (including opioids, NSAIDS, and antibiotics) pre- and post-surgery

[Methods, paragraph 11]

44. RESULTS 

1. Missing data for alcohol consumption of 17% is high. Should do sensitivity analysis without that variable otherwise justify why that is not necessary. 

We have carried out a sensitivity analysis, re-fitting the final model excluding alcohol consumption, and the findings remain unchanged.

45. 2. Clarify you checked for correlations between medication use pre and post-op. Same with opioid use and pain pre and post-op? There may be collinearity there. 

We have checked for evidence of multicollinearity using variance inflation factors as part of our model regression diagnostics, and found no evidence of this being an issue. See above point 8.

46. Table 1

Clarify that ‘complication’ is different to say surgery for MUA? I would argue MUA is a subset of complication (as this applies to some other complications too). By keeping them separate, this assumes different complications have different ‘weights’ so to speak. Can you justify/explain this approach. Seems like you have distinguished ‘medical’ from surgical and surgical is broken down further? 

In advance, we chose to separate out manipulation under anaesthetic, as this is treated as an operation using OPCS4 codes. Medical and surgical complications were considered as separate predictors. There was no specific reason for the approach we chose, and the reviewer is correct that others may have defined this differently. We have described this as a limitation in the discussion section.

Medical and surgical complications were considered as separate predictors a priori and others may have defined complications differently from this study. 

[Discussion, Strengths and limitations]

47. Fig 1 - Please justify exclusion of underweight BMI 

See above

48. DISCUSSION

1. Discussing use of opioids post-operatively as a risk factor along side pre-op factors is confusing. It makes sense that opioid use is assoc with persistent pain if the pain is driving use. On the other hand, pre-op BMI as a predictor is completely different. It may be a predictor as opposed to opioid use which may not ‘predict’, but rather be reactionary. Can the authors try to tease this out or dela with this better. 

We have tried to clarify our meaning regarding opioid use: 

We have identified a number of risk factors that are associated with an increased risk of poor pain outcome. The strongest pre-operative risk factors were: having only mild knee pain symptoms at the time of surgery, being a current smoker, obesity, and living in the most deprived areas. Opioid and antidepressant medication use were also associated with worse pain outcomes. The strongest post-operative factors were revision surgery and manipulation under anaesthetic within three months after the operation. We identified a range of other important risk factors with more moderate effects in terms of absolute risk differences in pain outcome, including a history of previous knee arthroscopy, and use of opioids within the three months after surgery, in addition to a number of other risk factors. Those with the least pre-operative pain were more likely to move to a worse post-operative pain state and were most likely to take pain relieving medicines both pre- and post-surgery, including opioids.

[Discussion, Main findings]

We hope our responses are satisfactory, but please do not hesitate to contact us if more information is required. 

Yours sincerely

Professor Andy Judge

---

## [Decision Letter · Decision Letter 1]

26 Nov 2021

PONE-D-21-27576R1Risk factors associated with poor pain outcomes following primary knee replacement surgery:analysis of data from the Clinical Practice Research Datalink, Hospital Episode Statistics and Patient Reported Outcomes as part of the STAR research programmePLOS ONE

Dear Dr. Judge,

Thank you for submitting your manuscript to PLOS ONE. After careful consideration, we feel that it has merit but does not fully meet PLOS ONE’s publication criteria as it currently stands. Therefore, we invite you to submit a revised version of the manuscript that addresses the points raised during the review process.

We look forward to receiving your revised manuscript.

Kind regards,

Armando Almeida

Academic Editor

PLOS ONE

Journal Requirements:

Reviewers' comments:

Reviewer's Responses to Questions

**Comments to the Author**

1. If the authors have adequately addressed your comments raised in a previous round of review and you feel that this manuscript is now acceptable for publication, you may indicate that here to bypass the “Comments to the Author” section, enter your conflict of interest statement in the “Confidential to Editor” section, and submit your "Accept" recommendation.

Reviewer #1: (No Response)

Reviewer #2: All comments have been addressed

Reviewer #3: All comments have been addressed

2. Is the manuscript technically sound, and do the data support the conclusions?

Reviewer #1: Yes

Reviewer #2: Yes

Reviewer #3: Yes

3. Has the statistical analysis been performed appropriately and rigorously? 

Reviewer #1: Yes

Reviewer #2: Yes

Reviewer #3: Yes

4. Have the authors made all data underlying the findings in their manuscript fully available?

Reviewer #1: Yes

Reviewer #2: Yes

Reviewer #3: Yes

5. Is the manuscript presented in an intelligible fashion and written in standard English?

Reviewer #1: Yes

Reviewer #2: Yes

Reviewer #3: Yes

6. Review Comments to the Author

Reviewer #1: I thank the authors for their answers to my previous comments. I have some additional suggestions that are detailed below.

INTRODUCTION

Page 3, first paragraph: “Many patients can expect to achieve reductions in knee pain and improvements in functional outcomes [3]. The percentage who experience ongoing pain is variable [4], with up to 20% experiencing knee pain that impacts their quality of life [5]. Patients who experience this kind of pain after surgery have not received the expected benefit and for some their pain is worse than it was before the operation [6, 7].” – In this paragraph, the time point at which these patients still experience pain is not clear. I can assume that it refers to chronic pain, but it should it should be stated.

Page 3, last paragraph: “There is limited research focusing solely on pain status”. In fact, there is a large body of research focusing on acute and chronic pain after surgery, including systematic reviews. I do not think that this sentence is an accurate depiction of the state of the art.

Page 3, last paragraph: “This is important given that up to 20% of patients will have long-term pain after surgery [5].” – This information is already stated in the previous paragraph.

Page 4: Please consider if the objective would be improved by stating that the aim was to “identify pre and postoperative risk factors”, since the authors present this strategy as a novelty.

RESULTS

Table 1: I appreciate the clarifications made by the authors concerning Table 1. However, it is my opinion that the information in the table is not very intuitive to understand. One suggestion would be to change the heading of the third column to “Proportion of patients with poor pain response with and without each risk factor”. And then change the “Yes” and “No” to “With” and “Without”.

Also, please consider if using the term “risk factor” is adequate in this context. Only the statistical analyses tell us if each characteristic is a risk factor or not (and not the descriptive data).

Since this table only presents descriptive information, do the authors believe that it is accurate to make claims about risk factors based on its information? Probably the most accurate way of stating the results would be, for example “The highest proportion of patients with poor pain outcomes were in the group of current smokers, males, people living in the most deprived areas and those with inflammatory arthritis.”

The rationale to select reference categories should be clear in the statistical analyses section, even if it is usual practice.

The information provided by the authors in the response letter concerning the analysis of strength of association should be stated in the statistical analysis section (We are examining the strengths of association and not arbitrary measures of statistical significance with cut offs of for example p<0.05 or the related concept of whether the confidence interval includes the null value.)

Reviewer #2: Comments addressed satisfactorily

Reviewer #3: The authors have done well to address all the reviewer comments. Some reanalysis has been undertaken and substantial amendments to the manuscript have been made

7. PLOS authors have the option to publish the peer review history of their article (what does this mean?). If published, this will include your full peer review and any attached files.

Reviewer #1: No

Reviewer #2: **Yes: **Ian Harris, Professor of Orthopaedic Surgery, UNSW Sydney

Reviewer #3: **Yes: **Justine M Naylor

---

## [Author Response · Author response to Decision Letter 1]

9 Dec 2021

Dear Dr Almeida,

Thank you for reviewing our manuscript at PLOS ONE as per your email dated 15 Oct 2021. We have considered these comments carefully, and our responses and amendments to the manuscript are reported below.

Reviewer #1: I thank the authors for their answers to my previous comments. I have some additional suggestions that are detailed below. 

Thank you, we are very grateful for your comments.

1. INTRODUCTION 

Page 3, first paragraph: “Many patients can expect to achieve reductions in knee pain and improvements in functional outcomes [3]. The percentage who experience ongoing pain is variable [4], with up to 20% experiencing knee pain that impacts their quality of life [5]. Patients who experience this kind of pain after surgery have not received the expected benefit and for some their pain is worse than it was before the operation [6, 7].” – In this paragraph, the time point at which these patients still experience pain is not clear. I can assume that it refers to chronic pain, but it should it should be stated. 

Thank you for your comments we agree and have amended these sentences. 

Many patients can expect to achieve reductions in knee pain and improvements in functional outcomes following surgery [3]. The percentage who experience ongoing chronic knee pain post-surgery is variable [4], with up to 20% experiencing knee pain that impacts their quality of life after 3 months post-op [5]. Patients who experience this kind of pain after surgery have not received the expected benefit and for some their pain is worse than it was before the operation [6, 7].

[Introduction, paragraph 2]

2. Page 3, last paragraph: “There is limited research focusing solely on pain status”. In fact, there is a large body of research focusing on acute and chronic pain after surgery, including systematic reviews. I do not think that this sentence is an accurate depiction of the state of the art. 

We agree and have amended the sentence, to clarify our meaning. 

Although previous research has explored predictors of outcomes of knee replacement [8], most studies have focused on total scores encompassing several domains (e.g. pain, stiffness and function) and fewer studies have focussed solely on pain status [7].

[Introduction, paragraph 2]

3. Page 3, last paragraph: “This is important given that up to 20% of patients will have long-term pain after surgery [5].” – This information is already stated in the previous paragraph. 

We agree and have removed the sentence. 

[Introduction, paragraph 2]

4. Page 4: Please consider if the objective would be improved by stating that the aim was to “identify pre and postoperative risk factors”, since the authors present this strategy as a novelty. 

We agree and have added this statement.

The aim of this study is to identify pre- and post-operative risk factors for whether or not a patient has a poor pain outcome after knee replacement surgery…

[Introduction, paragraph 3]

5. RESULTS

 Table 1: I appreciate the clarifications made by the authors concerning Table 1. However, it is my opinion that the information in the table is not very intuitive to understand. One suggestion would be to change the heading of the third column to “Proportion of patients with poor pain response with and without each risk factor”. And then change the “Yes” and “No” to “With” and “Without”. 

We agree and have amended the table headings.

Proportion of patients with poor pain response with and without each risk factor

Without With

[Table 1]

6. Also, please consider if using the term “risk factor” is adequate in this context. Only the statistical analyses tell us if each characteristic is a risk factor or not (and not the descriptive data). 

We agree and have clarified our wording for the descriptive data.

The aim of this study is to identify pre- and post-operative risk factors for whether or not a patient has a poor pain outcome after knee replacement surgery, by analysing a wide range of potential factors from the UK Clinical Practice Research Datalink (CPRD) primary care GOLD database linked to English Hospital Episode Statistics (HES) hospital admissions and to Patient Reported Outcomes Measures (PROMs) data.

[Introduction, paragraph 3]

Table 1. Descriptive statistics describing the total number of patients with each potential risk factor, and the proportion of patients with a poor pain outcome, according to whether or not they have the factor

[Table 1]

7. Since this table only presents descriptive information, do the authors believe that it is accurate to make claims about risk factors based on its information? Probably the most accurate way of stating the results would be, for example “The highest proportion of patients with poor pain outcomes were in the group of current smokers, males, people living in the most deprived areas and those with inflammatory arthritis.” 

We agree and have amended this sentence.

The highest proportion of patients with poor pain outcomes were in the group of current smokers, males, people living in the most deprived areas and those with inflammatory arthritis (Table 1).

[Results, paragraph 3]

8. The rationale to select reference categories should be clear in the statistical analyses section, even if it is usual practice. 

We agree and have added the following sentence.

As our dataset is large, we selected the lowest category for each variable as the reference category.

[Statistical analysis, paragraph 2]

9. The information provided by the authors in the response letter concerning the analysis of strength of association should be stated in the statistical analysis section (We are examining the strengths of association and not arbitrary measures of statistical significance with cut offs of for example p<0.05 or the related concept of whether the confidence interval includes the null value.) 

We agree and have added this sentence.

We examine the strength of associations and not arbitrary measures of statistical significance with cut offs of for example p<0.05 or the related concept of whether the confidence interval includes the null value.

[Statistical analysis, paragraph 2]

Reviewer #2: Comments addressed satisfactorily 

Thank you.

Reviewer #3: The authors have done well to address all the reviewer comments. Some reanalysis has been undertaken and substantial amendments to the manuscript have been made 

Thank you.

We hope our responses are satisfactory, but please do not hesitate to contact us if more information is required. 

Yours sincerely

Professor Andy Judge

---

## [Editor Report · Decision Letter 2]

13 Dec 2021

Risk factors associated with poor pain outcomes following primary knee replacement surgery:analysis of data from the Clinical Practice Research Datalink, Hospital Episode Statistics and Patient Reported Outcomes as part of the STAR research programme

PONE-D-21-27576R2

Dear Dr. Judge,

We’re pleased to inform you that your manuscript has been judged scientifically suitable for publication and will be formally accepted for publication once it meets all outstanding technical requirements.

Kind regards,

Armando Almeida

Academic Editor

PLOS ONE
---

## [Editor Report · Acceptance letter]

20 Dec 2021

PONE-D-21-27576R2 

Risk factors associated with poor pain outcomes following primary knee replacement surgery:analysis of data from the Clinical Practice Research Datalink, Hospital Episode Statistics and Patient Reported Outcomes as part of the STAR research programme 

Dear Dr. Judge:

I'm pleased to inform you that your manuscript has been deemed suitable for publication in PLOS ONE. Congratulations! Your manuscript is now with our production department. 

Kind regards, 

on behalf of

Prof. Armando Almeida 

Academic Editor

PLOS ONE